# Using Satellite Image Fusion to Evaluate the Impact of Land Use Changes on Ecosystem Services and Their Economic Values

Wang Shuangao [1,*], Rajchandar Padmanaban [2], Aires A. Mbanze [2,3,4], João M. N. Silva [2], Mohamed Shamsudeen [5,6], Pedro Cabral [7] and Felipe S. Campos [7]

[1] Shoujian Financial Center, Beijing Institute of Science and Technology Information, No. 140, Xizhimenwai Street, Xicheng District, Beijing 100044, China
[2] Forest Research Centre, School of Agriculture, University of Lisbon, 1349-017 Lisboa, Portugal; rajchandar@isa.ulisboa.pt (R.P.); ambanze@unilurio.ac.mz (A.A.M.); joaosilva@isa.ulisboa.pt (J.M.N.S.)
[3] Nova School of Business and Economics, Universidade Nova de Lisboa, 2775-405 Carcavelos, Portugal
[4] Faculty of Agrarian Sciences, Department of Environment and Nature Conservation, Universidade Lúrio, Sanga 3302, Mozambique
[5] Thoothukudi Campus, College of Engineering, Anna University, University V.O.C., Tamil Nadu 628008, India; shamsudeen@asprotechnologies.com
[6] Aspro Technologies, 33, 3rd Floor, Srirose Complex, Muthamil Street, Veepamoodu Junction, Nagercoil 629001, India
[7] NOVA Information Management School (NOVA IMS), Campus de Campolide, Universidade Nova de Lisboa, 1070-312 Lisbon, Portugal; pcabral@novaims.unl.pt (P.C.); fcampos@novaims.unl.pt (F.S.C.)
* Correspondence: h130031@e.ntu.edu.sg

**Abstract:** Accelerated land use change is a current challenge for environmental management worldwide. Given the urgent need to incorporate economic and ecological goals in landscape planning, cost-effective conservation strategies are required. In this study, we validated the benefit of fusing imagery from multiple sensors to assess the impact of landscape changes on ecosystem services (ES) and their economic values in the Long County, Shaanxi Province, China. We applied several landscape metrics to assess the local spatial configuration over 15 years (2004–2019) from fused imageries. Using Landsat-7 Enhanced Thematic Mapper Plus (ETM+), Landsat-8 Operational Land Imager (OLI) and Indian Remote Sensing Satellite System Linear Imaging Self Scanning Sensor 3 (IRS LISS 3) imageries fused for 2004, 2009, 2014 and 2019, we reclassified land use/land cover (LULC) changes, through the rotation forest (RF) machine-learning algorithm. We proposed an equivalent monetary metric for estimating the ES values, which also could be used in the whole China. Results showed that agriculture farmland and unused land decreased their spatial distribution over time, with an observed increase on woodland, grassland, water bodies and built-up area. Our findings suggested that the patterns of landscape uniformity and connectivity improved, while the distribution of landscape types stabilized, while the landscape diversity had a slight improvement. The overall ES values increased (4.34%) under a benefit transfer approach, mainly concerning woodland and grassland. A sensitivity analysis showed the selected economic value (EV) was relevant and suitable for the study area associated with our ES for LULC changes. We suggested that changes in landscape patterns affected the ESV trends, while the increases on some LULC classes slightly improved the landscape diversity. Using an interdisciplinary approach, we recommend that local authorities and environmental practitioners should balance the economic benefits and ecological gains in different landscapes to achieve a sustainable development from local to regional scales.

**Keywords:** landscape patterns; urban ecosystem services; environmental monitoring; remote sensing; image fusion

## 1. Introduction

Anthropic pressure on human-induced landscapes is the main driver of land use/land cover (LULC) changes and its effects on ecosystem services (ES) [1,2] which is key to



mitigate negative impacts for improved conservation outcomes. The use of ES has been proposed to define important contributions of ecosystems to human well-being, representing a link between biodiversity conservation and development needs [3]. However, ecological disturbances can affect several landscape patterns and their role in the provisioning of indispensable goods and services, such as potable water, non-timber forest products (NTFPs), erosion protection and soil nutrition [4–6]. The relation between landscape change and ES is mutually dependent. For instance, land use changes can lead to strong or slight alterations in ecosystem components, structures, ecological processes and biodiversity patterns [3,7]. On the other hand, the degradation and losses on the ES also affect the structure and aesthetic of landscape, with serious consequences for human safety and health, and directly threaten regional and even global ecological integrity [8]. Therefore, studying the changes of landscape patterns on ES can effectively grasp the changing trend of regional ecological environment, rationally allocating land use activities to promote harmonious and sustainable development goals for human and nature [9,10].

The inclusion of economic costs in ES assessments is an important premise of ecological valuation to support environmental compensation policies [9], which could improve global and national green development accounts [11]. Globally, quantitative analyses of ES valuation suggest a need for the development of land use plans that optimally balance economic costs and ecological constraints in fast-developing countries [12–14]. China is one of the fast-growth emerging nations in the world [2], achieving its higher record of gross domestic product (GDP) development of 9.6% in the last 20 years [15,16]. However, this relative fast growth was made at the expense of the ES depletion and high environmental damage [17–19]. With increasing negative environmental externality that affected the wellbeing and the public health of their local citizens [15], the Chinese government decided to embrace an ambitious plane of sustainable development that required changes in key sectors such as agriculture, energy and industry [6,15,16,20,21]. As a result, there have been a growing number of studies attempting to measure landscape enhancement and the value of the different ES [8–10,14,22]. Yet, only a few studies address the impact of those programs on landscape recovering [23,24], and the ecological values arise from the enhancement of ES [4]. The correctness of this assessment technique is still low and does not conform to China's national conditions [25]. Regarding the research results of foreign countries, made some necessary improvements to analyze the dynamic changes of China's regional ES and estimate its economic value. However, this method was never being ascertained in small-scale areas out of the Chinese countrywide.

Remote sensing (RS) provides many approaches to evaluate LULC changes in urban and rural landscapes, as well as for estimating the socio-economic impacts of ES [26]. Available RS approaches determine the benefits for LULC monitoring and calculating economic values on ES over conventional techniques that are based on field investigation combined with single sensor satellite images or aerial photographs [15]. Therefore, RS practices have been extensively applied for estimating LULC changes associated with economic impacts on ES in several countries worldwide, such as Germany, USA, Canada, and India [13,14,27]. Predominantly, the advantage of combining/fusing data from multiple sensors with varied spectral and spatial resolution provides more detailed information than the single sensor analysis [26]. Fused imagery from multiple sensors deliver significant metrics for the estimation of LULC change dynamics in an urban area as well the quantification of the relationship between variations in ES and their economic values [27,28].

In this research, we evaluate land use changes on ES and their EV in Long County, China, from 15 years (2004–2019) using RS images. We select this area because it is an agricultural county in a poor mountainous area of the Shaanxi Province with a fragile ecological environment [29–31], where the growth of human population and urbanization has been exacerbating the existing scarce natural resources [29]. We aim to explore how land use changes can influence environmental policies implementation for regional sustainable development, focused on ES enhancement and protection, landscape dynamics and resource allocation in the region. To achieve this aim, we develop an integrative spatial

approach with the following specific objectives: (i) to compute 15 years of LULC change dynamics in the Long County using fused imageries; (ii) to quantify the landscape patterns and characteristics of its structure and spatial configuration using landscape metrics; (iii) to assess ES value based on the ecological value estimation method; and (iv) to verify if the selected economic value is relevant and suitable for the study area using the sensitivity analysis. This paper is organized as follows. Sections 2.1 and 2.2 describes the study area and data used in this work. Section 2.3 summarize the research design of this paper. Section 2.4 provides information about satellite data preprocessing followed. Section 2.5 describes the image fusion method. Section 2.6 provides step by step rotation forest classification algorithm to produce LULC maps. Section 2.7 explains different method followed in this project for landscape pattern evaluation. Section 2.8 explains the method used in this work to assess ecosystem services value. Section 2.9 explains the sensitivity analysis. Section 3 then combines the results from the individual stages of the research design to evaluate land use and land cover changes on ecosystem services and their economic values. Finally, Sections 4 and 5 provide the discussion and conclusions.

## 2. Materials and Methods

### 2.1. Study Area

The study is focused on Long County, located in the west of Guanzhong Plain and the northwest of Baoji City, China (Figure 1). Long County is 57.6 km wide from north to south, 59.7 km long from east to west [32]. The climate of the region is sub-humid continental monsoon with an average temperature of 10.9 °C, a frost-free period is 197 days [31]. The average yearly precipitation is 677.1 mm [31].

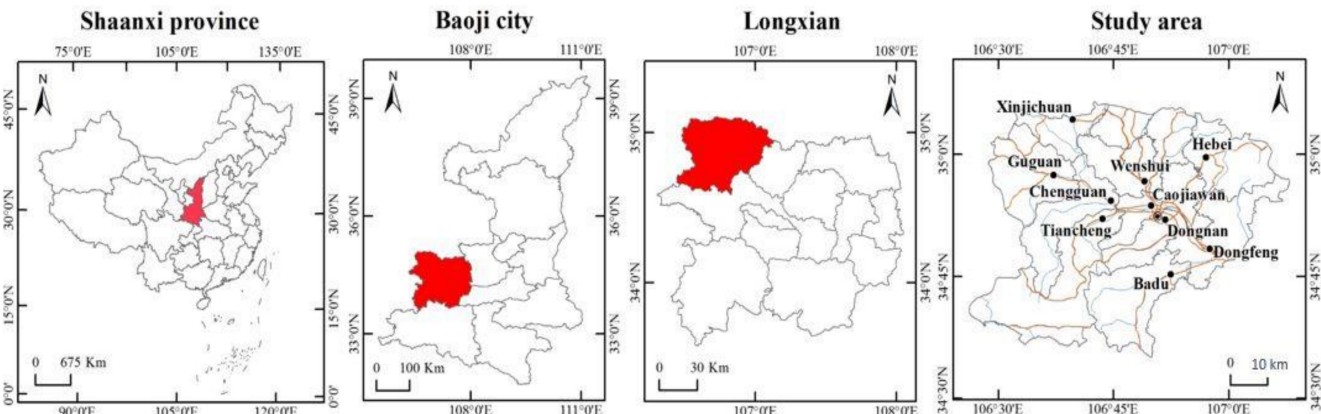

**Figure 1.** Study area covering the Longxian (Long) County, Baoji City, Shaanxi Province, China.

In 2019, Longxian (Long) County governs 10 towns, with a population of 252,000, realizing a regional GDP of 9.155 billion yuan, including 1.863 billion yuan of the added value of the primary industry, 3.763 billion yuan of added value of the secondary industry, and 3.529 billion yuan of the tertiary industry. Based on the permanent population, the per capita GDP is 36331 yuan [32].

### 2.2. Data

We used level-1 satellite imagery covering the study area from Landsat-7 Enhanced Thematic Mapper (ETM+) with 8 bands and Landsat-8 Operational Land Imager (OLI) sensors with 11 bands [33], encompassing the years of 2004, 2009, 2014 and 2019 (Table 1). We used the cloud-free Landsat data downloaded from the United States Geological Survey (USGS) portal, in GeoTIFF format. We used GeoTiff format Indian Remote Sensing Satellite System Linear Imaging Self Scanning Sensor 3 (IRS LISS III) satellite imageries with 4 bands downloaded from the National Remote Sensing Centre (NRSC), Indian Space Research Organisation (ISRO). To standardize the coordinate reference system, we projected the

satellite images to the Universal Transverse Mercator (UTM), using the World Geodetic System (WGS) 1984 datum.

**Table 1.** Landsat Enhanced Thematic Mapper (ETM+), Operational Land Imager (OLI) imageries and Linear Imaging Self Scanning Sensor- 3 (LISS III) used in this study.

| Date of Acquisition | Sensor Used | Spatial Resolution |
|---|---|---|
| 06 June 2004 | Landsat-7 ETM+ | 30 m |
| 03 June 2009 | Landsat-7 ETM+ | 30 m |
| 12 June 2014 | Landsat-8 OLI | 30 m |
| 03 June 2019 | Landsat-8 OLI | 30 m |
| 23 June 2004 | LISS–III | 23.5 m |
| 21 June 2009 | LISS–III | 23.5 m |
| 26 June 2014 | LISS–III | 23.5 m |
| 23 June 2019 | LISS–III | 23.5 m |

### 2.3. Research Design

To summarize our research design, we provided a schematic representation for the methodological framework used in this study (Figure 2), which is described in the following subsections.

### 2.4. Satellite Data Pre-Processing

To assess the LULC dynamics in the study area, we followed the pre-processing technique (Figure 2) for the Landsat and IRS-LISS III imagery using geometrically corrections through the "georef" and "geoshift" functions in the "Landsat" package in R software [34] using ground control points (GCP) from USGS (https://landsat.usgs.gov/gcp, accessed on 26 January 2019). The Landsat 7 Scan Line Corrector (SLC) has failed from 2003 [35]. Thus, data from 2004 and 2009 have data gaps, but are still beneficial and uphold the same geometric and radiometric corrections [35]. Therefore, we filled some missing data gaps (pixels) due to the scan line error happened in the ETM+ sensor in 2003, with the Landsat 7 Scan Line Corrector (SLC)-off Gap function [35,36]. After that, we rectified the SLC-off images by mosaicking according to the USGS gateway, filling the residual gaps through histogram correction [35]. We performed scan line error correction using the Earth Resource Development Assessment System (ERDAS), version 16.5 [37,38]. Using the Landsat ETM+ radiometric calibration of top-of-atmosphere (TOA) radiance [39], we transformed the digital number (DN) integer values (0–255) of the raster data to at-satellite radiance values. In addition, we used an atmospheric correction to verify the disparity between surface reflectance and at-sensor reflectance [39]. We identify clouds, aerosol and cirrus, with dark object and modified dark object subtraction methods [40]. To ensure the homogeneousness of reflectance values for the examination of vegetation dynamics, we used invariant features in images across 2004–2019 through the pseudo-invariant features (PIF) function with a major axis regression [41,42]. We conducted all the atmospheric corrections and radiometric corrections in R software [43].

### 2.5. Image Fusion

The preprocessed Landsat-7/Landsat-8 and LISS III satellite sensor images were fused using the "Ehlers" image fusion technique to overcome the problem of spectral changes in agricultural areas and suburban lands [44]. Commonly, earth observation satellites sense the data as high-resolution panchromatic and low-resolution multispectral so image fusion is used to make use of better decisions [45]. Ehlers fusion helps us to preserve the spectral characteristics with minimal color distortion while doing multi-sensor and multi-data fusion [46]. It works by intensity–hue–saturation (IHS) transform which separates spectral and spatial information in a standard Red Green Blue (RGB) image [47]. We fused the high-resolution spatial structure from LISS III with the low-resolution spectral information from Landsat to construct a high-resolution multispectral image for land use and landcover

classification. The idea of fusion algorithms is to sharpen the remotely sensed data by enhancing edges and grey level discontinuities without making changes on multispectral components in homogeneous areas [48]. Two things need to be taken care of to facilitate these requirements, as (i) optimal colour and spatial details must be separated and (ii) spatial information should be manipulated to allow adaptive enhancement of images [49].

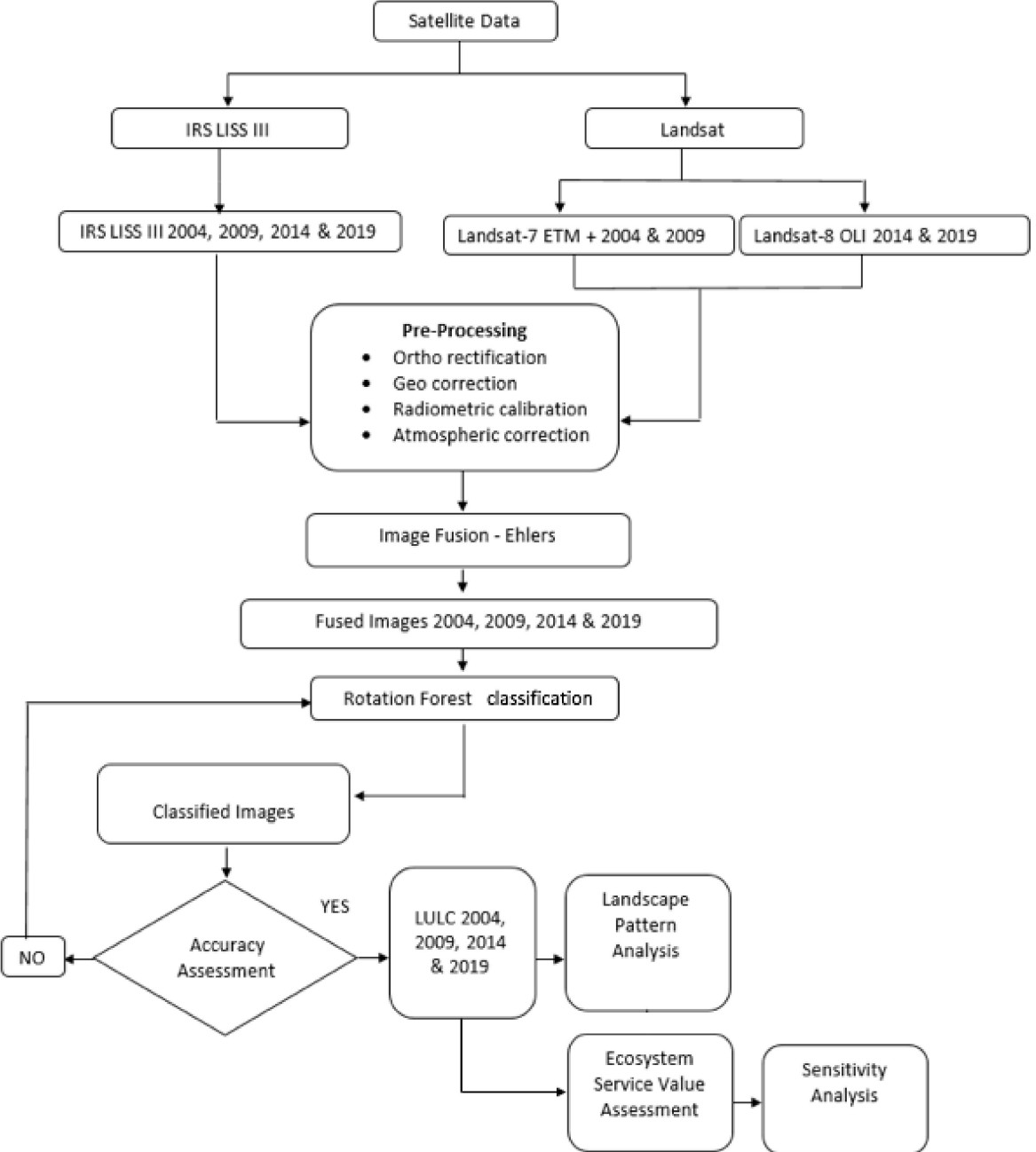

**Figure 2.** Methodology for analyzing ecosystem services and their economic values from land use and land cover changes.

The IHS transform is used for optimal colour separation similarly when the number of bands available is three, multiple IHS transform is employed until the number of bands available [48]. Since the spectral characteristics of every band varies, the order of bands must be specified for IHS transform [44]. Adaptive filter design can be obtained by the intensity component of the Fourier transform and the panchromatic image in the frequency domain. Spatial components are enhanced or suppressed using fast Fourier transform (FFT) [50]. Low pass filter is used for enhancing the intensity spectrum whereas high pass

filter for enhancing the spectrum of high-resolution image [44]. Once filters are applied, inverse FFT is used to bring image back to spatial domain and fused to form new intensity component and consists of low frequency information from the low-resolution multi-spectral image and high frequency information from the high-resolution data generally panchromatic image [45]. A new IHS image is formed with the new intensity component and the hue and saturation component of the original multispectral image. Finally, a fused RGB image is obtained with spatial characteristics from panchromatic image and spectral characteristics from multispectral image by applying inverse IHS transform [44]. These steps are repeated until all the given bands are fused with panchromatic image. The order of a spectral bands in multiple IHS transform is not important because colour preservation procedures are employed in this fusion.

We used the "panSharpen" function of the "RStoolbox" package in R for the image fusion [26]. To avoid color distortion and damage of terrain feature information, we maintained all bands from Landsat-7/Landsat-8 and LISS III for the combining process [13]. We retained 23.5 m resolution in fused images for further classification and spatial metrics calculations.

### 2.6. LULC Classification

We divided the landscape into six LULC classes (woodland, grassland, farmland, built-up area, water and unused land; Table 2), using the rotation forest (RF) machine learning algorithm, based on the ensemble construction technique to acquire enhanced predictive performance (Figure 2) with less number of trees [26,51]. The RF algorithm is often connected with decision tree (DT) methods, where each LULC class should be individually reconstructed for image processing [51], through the following steps: (i) K subset is randomly split from the feature set; (ii) a principal component analysis (PCA) is applied in each of the subsets to find the variability information; (iii) the indeterminate occurrences are reclassified; (iv) the regular buoyancy is computed for each class through classifiers; and (v) the label of each classified class is reallocated to the one with the maximum buoyancy value. We perform this reclassification method in R software (R Development Core Team, 2019).

**Table 2.** Description of land use/land cover (LULC) classes.

|   | LULC Classes | Land Uses Comprised in the IDLULC |
|---|---|---|
| 1 | Built-up area | Roads, man-made structures, and urban areas |
| 2 | Woodland | Dense vegetation, forest and timberland |
| 3 | Farmland | Agriculture and productive lands |
| 4 | Unused land | Drylands, non-productive lands and non-irrigated |
| 5 | Water bodies | Rivers, streams, lakes, open water, and ponds |
| 6 | Grassland | Grazing area, bushes and shrubbery |

We calculated and compared the accuracy of LULC between the non-fused and fused images of 2004, 2009, 2014 and 2019. We used Google Earth images from 2004, 2009, 2014 and 2019 of the Long country attained from Google Earth Engine (GEE) gateway [52] as a reference for the accuracy assessment of LULC maps from fused and non-fused images. We generated a group of random location points (25 for each class) and attained those values for 2004, 2009, 2014 and 2019. Then, the extracted random values from GEE images were compared to the classified LULC maps of 2004, 2009, 2014 and 2019. To examine the accuracy of classified LULC maps from fused and non-fused images, we employed the kappa coefficient using ERDAS Imagine 2018 (version 16.5). The producer and user accuracies were also computed using the confusion matrix classifier [53,54]. A kappa coefficient higher than 0.7 specifies an acceptable accuracy of classified LULC maps [54]. The accuracy rate is more than 85%, also verified by actual field sampling.

*2.7. Evaluation of Landscape Pattern*

2.7.1. Selection of Landscape Metrics

Landscape metrics is a quantitative index that can highly concentrate landscape pattern information and reflect characteristics of its structural composition and spatial configuration [25,55,56]. We used eleven landscape metrics to examine the landscape pattern information and characteristics of its structure and spatial configuration between 2004 and 2019 in Long County. Landscape metrics refers to a simple quantitative index that can provide very denser landscape pattern evidence at the patch level, landscape-level and class-area level, which is suitable for quantitative expression of the relationship between landscape pattern and ecological process [25]. To examine the classification patterns of LULC, we selected the landscape metrics: patch type area (CA), patch area ratio (PLAND), number of patches (NP), clumpiness index (CLUMPY), average patch area (AREA_MN), patch density (PD), landscape shape index (LSI), and largest patch index (LPI). For illustrating different types of classes, complexity and spatial characteristics of fragmentation, we also used the landscape-level patch number (NP), Shannon diversity index (SHDI), landscape shape index (LSI), patch richness (PR), Shannon evenness index (SHEI), contagion index (CONTAG), and average patch area (AREA_MN) [57,58] (see Table 3 for details). The different landscape metrics were computed in the FRAGSTATS software package (Version 4.2).

**Table 3.** Spatial metrics calculated to examine the landscape pattern information [55,57,58].

| Landscape Metrics | Formulas | Explanation | Values Range |
|---|---|---|---|
| Patch type area | $CA = \sum_{j=1}^{n} a_{ij} \left( \frac{1}{10000} \right)$ <br> $a_{ij}$ = area measures in m$^2$ of patch covering *ij*. | To quantify the class area in the landscape | CA > 0 |
| Patch area ratio | $PLAND = P_i = \frac{\sum_{j=1}^{n} a_{ij}}{100} (100)$ <br> $P_i$ = total landscape occupied by different patch. <br> $a_{ij}$ = area measures in m$^2$ of patch covering *ij*. | To quantifies landscape patch region ratio | $0 < PLAND \leq 100$ |
| Number of patches | $NP = n_i$ <br> $n_i$ = total number of patches in the region of patch type *i*. | To measure the total number of different patches of LULC | NP ≥ 1 |
| Landscape shape Index | $LSI = \frac{e_i}{min\ e_i}$ <br> $e_i$ = length of the different edges | To measure class aggregation for different class area | LSI 1 ≥ 1, without limit |
| Clumpiness Index | $Clumpy = [(G_i - P_i)/P_i$ <br> $for\ G_i < P_i\ \&\ P_i < 5,\ else$ <br> $G_i - P_i/1 - P_I]$ <br> $g_{ii}$ = total number of similar connections among pixels, *i* based doubled progression and $g_{ik}$ = total number of similar connections among pixels, *k* based doubled progression <br> $P_i$ = total landscape occupied by different patch. | To quantity the clumpiness of different patches in the urban area. Clumpiness shows the frequency with which various pairs of patch types appear side-by-side on the map | $-1 \leq CLUMPY \leq 1$ |
| Path Density | $PD = \frac{n_i}{A} (10,000)$ <br> $n_i$ = total number of patches in the region of patch type *i*. <br> $A$ = total area in the landscape measures in m$^2$ | To calculate number of patches of equivalent patch type by total region | PD > 0 |

**Table 3.** *Cont.*

| Landscape Metrics | Formulas | Explanation | Values Range |
|---|---|---|---|
| Largest Patch Index | $LPI = \frac{max_{j=1}^{n}(a_{ji})}{A}(100)$<br>$a_{ij}$ = area measures in m$^2$ of patch covering $ij$<br>$A$ = total area in the landscape measures in m$^2$ | To measure the proportion of the landscape comprised by the major patch | $0 < LIP \leq 100$ |
| Average Patch area | $MN = \frac{\sum_{j=1}^{n} X_{ij}}{n_i}$<br>$n_i$ = total number of patches in the region of patch type $i$. | To examine the average area of the different patches | $0 < MN \leq 100$ |
| Shannon evenness index | $SHEI = \sum_{i}^{m}(P_i * \ln(P_i)) / \ln(m)$<br>$P_i$ = total landscape occupied by different patch.<br>$m$ = total number of patch classes | To provides information on area richness and composition | $0 \leq SHEI \leq 1$ |
| Shannon's diversity index | $SHDI = \sum_{i=1}^{m} (P_i \ln(P_i))$<br>$P_i$ = total landscape occupied by different patch.<br>$m$ = total number of patch classes | To provides information on diversity | $SHDI \geq 1$ |
| Contagion index | $Contag = [1 + \sum_{i=1}^{m} \sum_{k=1}^{m}[(p_i)$<br>$\{g_{ik} / \sum_{k=1}^{m} g_{ik}\}$<br>$\{\ln(p_i)[g_{ik} / \sum_{k=1}^{m} g_{ik}] / 2\ln(m)]100$<br>$g_{ik}$ = total number of similar connections among pixels, $k$ based doubled progression<br>$P_i$ = total landscape occupied by different patch.<br>$m$ = total number of patch classes | To calculate the heterogeneity | Percent < Contagion $\leq 100$ |

### 2.7.2. Landuse Use Degree

We also calculated the land use degree (LUD) to analyze the extent and complexity of regional land use, which can quantitatively reflect its natural attributes and the comprehensive consequence of human disturbance [4,59]. $L_u$ is mathematically defined as:

$$L_u = 100 \times \sum_{i=1}^{n} P_i \times Q_i \qquad (1)$$

where $L_u$ is the comprehensive index of land use degree; $P_i$ is the grade I land use degree grading index; $Q_i$ is the percentage of grade I land use grade area.

### 2.8. Assessment of Ecosystem Services Values

The assessment of ecosystem service values (ESV) is primarily based on the economic estimation for different types of ecosystems under a benefit transfer approach [60]. In this research, we propose an equivalent metric for estimating the value of ecosystem services [61], which also could be used in the whole of China. The calculation formula of the ESV coefficient [61] is as follows:

$$VC_0 = \frac{1}{7} \times P \times \frac{1}{n} \sum_{i=1}^{n} Q_i \qquad (2)$$

where $VC_0$ is the value of ESV equivalent factor (yuan hm$^{-2}$a$^{-1}$), (a is the equivalent factor for every year), yuan is the monetary unit of China (¥); $P$ is the average grain price (yuan × kg$^{-1}$) (different in each year); $Q$ is the average grain yield (kg × hm$^{-2}$) and n is the number of years. The value equivalent multiplied by the economic value of the natural grain output of the productive land quantifies the ecological value per unit area of the different landscape type. The formula for evaluating ESV [30] in Long County is:

$$EVS = \sum_{k=1}^{n} (A_k \times VC_k) \tag{3}$$

where *ESV* measures in yuan; $A_k$ is the area in hm$^2$ of landscape type *K*, and $VC_k$ is the *ESV* coefficient (yuan hm$^{-2}$ a$^{-1}$).

For each land cover type, the services delivered by the ecosystem are identified and given a monetary value based on Long County socioeconomic situations as well as supply and demand. The assessed per square hectometer value of every ecosystem is then multiplied by the area of each biome to estimate the sum of the total monetary value of the Long County ecosystem [61].

*2.9. Sensitivity Analysis*

To verify whether the selected economic value (VC) was suitable for the study area, we used the standard economic elasticity concept to calculate the coefficient of sensitivity (CS) [62]. Sensitivity analysis helps to identify the impact of uncertainty in the coefficient values after obtaining the optimal solution of the model [62]. Sensitivity analysis is useful to regulate the dependence level of the change of the ESV upon the coefficient value [63].

The dependence of ESV and VC on time change can be determined by CS, which means that 1% change in VC cause 1% variation in ESV [62]. If CS > 1, it indicates that the ESV is changeable to VC [61,64].; CS < 1 specifies that the ESV is unchangeable to VC, thus VC is introduced appropriately, and the results are reliable [64]. The CS formula is as follows [63]:

$$CS = \frac{(ESV_j - ESV_i)/ESV_i}{(VC_{jk} - VC_{jk})/VC_{ik}} \tag{4}$$

where *CS* is the sensitivity, refers to the change of ESV produced by 1% variation of *VC*. The *VC* is the ecological value coefficient; *i* and *j* are the initial value coefficients and the adjusted coefficients; *k* stands for different land uses [63].

## 3. Results

*3.1. Landscape Type Changes*

The LULC maps from 2004 to 2019 showed that farmland decreased their spatial distribution over time ($-42703$ hm$^2$) (Figure 3), reaching the pick in 2019. Grassland first increased and then decreased, with an overall change of 317 hm$^2$. Woodland also followed a similar pattern, with an overall change of 42,199 hm$^2$, whereas the built-up area increased 1487 hm$^2$. In general, the spatial distribution of water bodies also increased with a final gain of 906 hm$^2$. The overall unused land decreased, and the weakening trend slowed down in 2004, with an overall change of $-1717$ hm$^2$.

We attained kappa coefficient values of 0.86, 0.88, 0.85 and 0.86 with an overall accuracy value of 87%, 89%, 86% and 87% for the LULC classification for 2004, 2009, 2014 and 2019 respectively using fused images (Table 4). In contrast, the average kappa coefficient and overall accuracy values for LULC classification using non-fused images for 2004, 2009, 2014 and 2019 were considerably lower, i.e., 0.77, 0.74, 0.74 and 0.74 and 76%, 75%, 75%, and 75%, respectively. Hence, the accuracy of LULC classification using non-fused images was significantly lower than the LULC classification using fused images. The accuracy of the LULC from the fused images is considerably higher because of the higher spatial resolution and number of bands available in fused images than non-fused images. Therefore, we used the LULC from fused images to calculate the ecosystem service and their economic values.

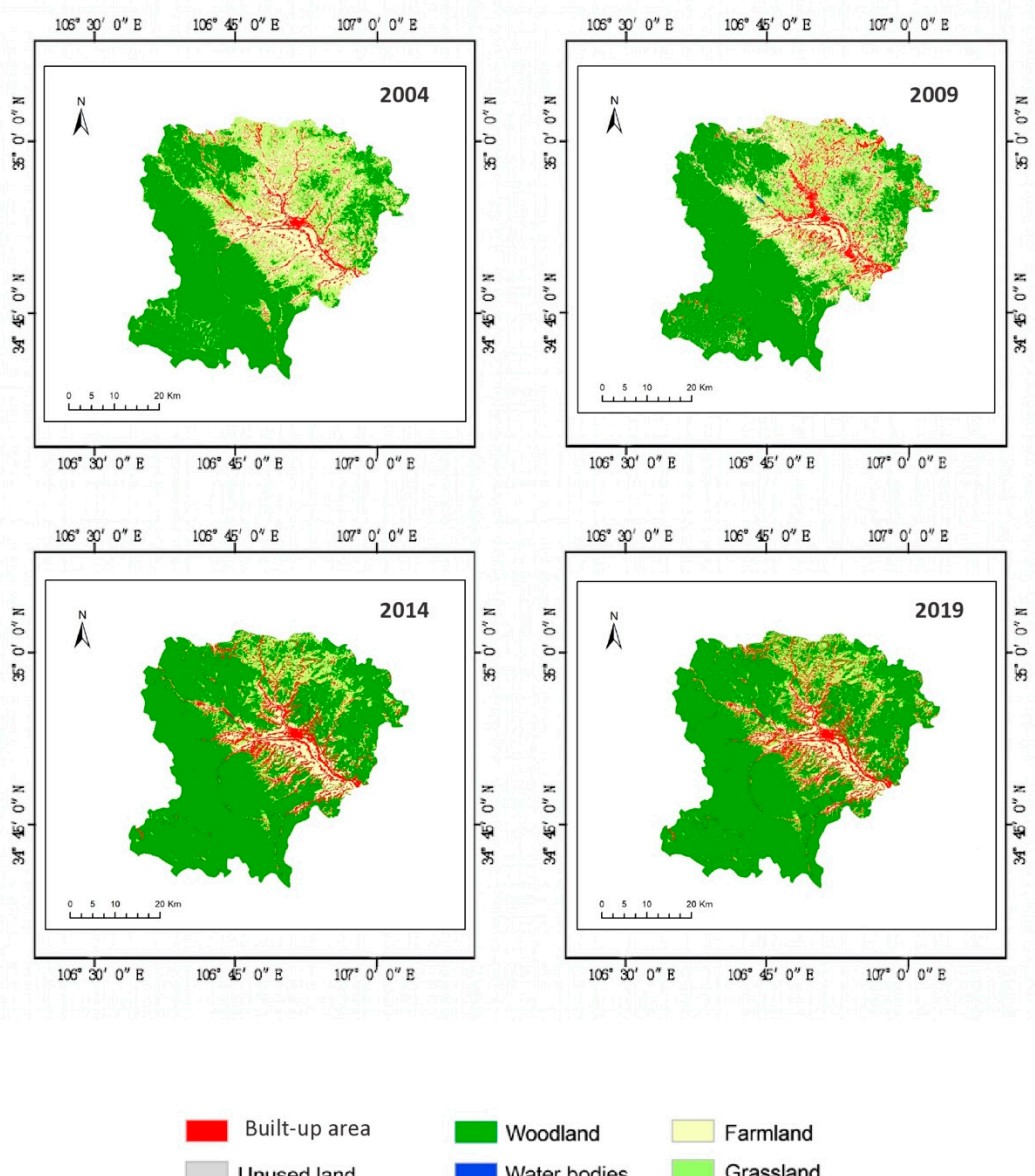

**Figure 3.** LULC change during the 15 years of the study period (2004–2019) in Long County, Shaanxi Province, China.

**Table 4.** Accuracy assessment values for the LULC using non-fused and fused images by 2004, 2009, 2014 and 2019. Producer accuracy = PA; user accuracy = UA.

| | 2004 | | | | 2009 | | | | 2014 | | | | 2019 | | | |
|---|---|---|---|---|---|---|---|---|---|---|---|---|---|---|---|---|
| | Non-Fused | | Fused | | Non-Fused | | Fused | | Non-Fused | | Fused | | Non-Fused | | Fused | |
| **Classes** | **PA** | **UA** | **PA** | **UA** | **PA** | **UA** | **PA** | **UA** | **PA** | **UA** | **PA** | **UA** | **PA** | **UA** | **PA** | **UA** |
| Built-up area | 77.1 | 76.3 | 81.7 | 82.4 | 70.5 | 71.3 | 89.3 | 86.2 | 76.4 | 78.2 | 88.1 | 84.3 | 71.1 | 74.0 | 87.6 | 83.9 |
| Woodland | 76.5 | 78.2 | 87.2 | 88.1 | 71.3 | 73.9 | 89.0 | 86.4 | 72.8 | 76.5 | 90.3 | 90.6 | 72.9 | 76.3 | 89.1 | 90.8 |
| Farmland | 71.4 | 77.5 | 89.2 | 92.3 | 73.4 | 74.1 | 87.8 | 89.1 | 76.1 | 77.9 | 90.8 | 92.5 | 75.1 | 77.3 | 87.2 | 88.5 |
| Unused land | 76.1 | 78.9 | 90.1 | 93.3 | 76.5 | 77.9 | 89.1 | 92.3 | 71.8 | 73.4 | 87.6 | 91.2 | 72.7 | 77.9 | 89.3 | 90.4 |
| Water bodies | 78.7 | 79.8 | 87.6 | 88.9 | 77.1 | 75.4 | 90.1 | 92.3 | 76.3 | 79.5 | 80.1 | 81.5 | 80.1 | 81.8 | 82.4 | 85.9 |
| Grassland | 80.1 | 81.2 | 90.6 | 94.5 | 82.4 | 83.1 | 92.6 | 93.1 | 80.4 | 81.5 | 82.3 | 83.8 | 80.5 | 80.9 | 89.9 | 93.2 |
| Overall Accuracy | 76.5 | | 87.7 | | 75.2 | | 89.6 | | 75.6 | | 86.5 | | 75.4 | | 87.5 | |
| Kappa | 0.77 | | 0.86 | | 0.74 | | 0.88 | | 0.74 | | 0.85 | | 0.74 | | 0.86 | |

We obtained a transition matrix aiming to provide a quantitative description of the transitions in LULC between 2004 and 2019 for all estimated classes (Table 5). The transition matrix of the landscape area from 2004 to 2019 is presented in Table 6. It can be observed that most of the changes were transformed to farmland accounting for 86.45% of the total converted area. This spatial transformation trend was followed by woodland (7%), unused land (3.12%), built-up area (2.58%), and grassland (0.79%). The rest of the LULC types were marginally converted.

**Table 5.** Land area change and its respective percentage of each class of LULC during the study period (2004–2019).

| Landscape Type | 2004 | | 2009 | | 2014 | | 2019 | | Change in Area (hm$^2$) | | | |
|---|---|---|---|---|---|---|---|---|---|---|---|---|
| | Area (hm$^2$) | % | Area (hm$^2$) | % | Area (hm$^2$) | % | Area (hm$^2$) | % | 2004–2009 | 2009–2014 | 2014–2019 | 2004–2019 |
| Farmland | 94,627 | 41.25 | 74,865 | 32.85 | 63,840 | 28.01 | 51,924 | 22.78 | −19762 | −11,025 | −11,916 | −42,703 |
| Grassland | 481 | 0.21 | 1884 | 0.83 | 374 | 0.16 | 164 | 0.07 | 1403 | −1510 | −210 | −317 |
| Woodland | 127,941 | 56.14 | 147,757 | 64.84 | 158,739 | 69.66 | 170,140 | 74.66 | 1,9816 | 10,982 | 11,401 | 42,199 |
| Built-up area | 2989 | 1.31 | 2964 | 1.30 | 4271 | 1.87 | 4476 | 1.96 | −25 | 1307 | 205 | 1487 |
| Water bodies | 104 | 0.05 | 376 | 0.16 | 248 | 0.11 | 1154 | 0.51 | 272 | −128 | 906 | 1050 |
| Unused land | 1752 | 0.77 | 47 | 0.02 | 421 | 0.18 | 35 | 0.02 | −1705 | 374 | −386 | −1717 |
| Total | 227,894 | 100 | 227,893 | 100 | 227,893 | 100 | 227,893 | 100 | - | - | - | - |

**Table 6.** Landscape type area transition matrix from 2004 to 2019.

| Landscape Types in 2004 | Landscape Types in 2019 | | | | | | |
|---|---|---|---|---|---|---|---|
| | Farm Land hm$^2$ | Grass Land hm$^2$ | Wood Land hm$^2$ | Built-Up Area hm$^2$ | Water Bodies hm$^2$ | Unused Land hm$^2$ | Decreased Ratio % |
| Farm land | 512,109 | 60 | 503,241 | 28,041 | 7814 | 142 | 86.45 |
| Grass land | 11,21 | 390 | 3765 | 60 | 5 | 1 | 0.79 |
| Woodland | 38,564 | 1366 | 1,377,881 | 1602 | 1906 | 243 | 7.00 |
| Built-up area | 12,989 | - | 1537 | 17,097 | 1582 | 3 | 2.58 |
| Water bodies | 108 | - | 58 | 122 | 868 | - | 0.05 |
| Unused land | 12,046 | 2 | 3959 | 2816 | 647 | - | 3.12 |
| New increased area (hm$^2$) | 64,828 | 1428 | 512,560 | 32,641 | 11,954 | 389 | - |
| Increased proportion (%) | 10.39 | 0.23 | 82.17 | 5.23 | 1.92 | 0.06 | 100 |

### 3.2. Estimation of LULC Changes

Aggregate results for all LULC changes during the 15 years evaluated (2004–2019) are shown in Table 7. The results indicated that overall, the number of patches, patch density and landscape shape index contracted in −1799, −0.7 and −1.1 respectively. In contrast, the maximum patch index, mean patch area and spread degree increased 9.01%, 1209hm$^2$ and 4 respectively. Landscape richness did not represent change, while Shannon diversity and uniformity slightly increased along the years.

**Table 7.** Aggregate changes on LULC in Long County during the 15 years (2004–2019), based on different landscape metrics.

| Year | Number of Patches | Patch Density | Maximum Patch Index | Landscape Shape Index | Mean Patch Area | Contagion Index | Patch Richness | Shannon Diversity Index | Shannon Evenness Index |
|---|---|---|---|---|---|---|---|---|---|
| | (NP) | (PD) | (LPI %) | (LSI) | (hm$^2$) | (Contag) | (PR) | (SHDI) | (SHEI) |
| 2004 | 28,712 | 12.5989 | 44.0108 | 74.8318 | 18,088.33 | 68.445 | 6 | 0.7998 | 0.44 |
| 2009 | 26,857 | 11.7849 | 37.9496 | 79.0558 | 19,337.64 | 69.3691 | 6 | 0.8011 | 0.44 |
| 2014 | 27,836 | 12.2145 | 48.5861 | 85.6635 | 18,657.60 | 70.0838 | 6 | 0.8197 | 0.45 |
| 2019 | 26,913 | 11.8095 | 53.028 | 73.7931 | 19,297.53 | 72.4943 | 6 | 0.8257 | 0.46 |

When the analysis was expanded for the different categories of land use (i.e., wood­land, farmland, built-up area, water, grassland, and unused land), the results showed an overall improvement for unused land, woodland, and grassland, while built-up area and farmland decreased their spatial representation. Indeed, the overall farmland for agricul­ture and unused land decreased, while woodland, grass and built-up area increased. The patch area ratio also followed a similar trend. In the half of the first decade (2004–2009), the landscape shape index declined for woodland (3.7%) and farmland (3.5%), while stabilized in the following years. Landscape shape index did not change for grassland during all the 15 years assessed, showing an increase for built-up area and a decrease for unused land, despite slight stabilization between 2004 and 2009 (Table 8).

**Table 8.** Change on LULC in Long County during the 15 years (2004–2019), based on different landscape metrics.

| Landscape Type | Year | Patch Type Area | Patch Area Ratio | Number of Patches | Patch Density | Max Patch Index | Landscape Shape Index | Mean Patch Area | Concentration |
|---|---|---|---|---|---|---|---|---|---|
| | | CA km² | PLAND % | NP | PD | LPI | LSI | MN km² | CLUMPY |
| Woodland | 2004 | 127,940.5 | 56.14 | 8477 | 3.72 | 44.01 | 78.72 | 34,395.21 | 0.85 |
| | 2009 | 147,756.8 | 64.84 | 9601 | 4.21 | 37.95 | 82.25 | 35,072.06 | 0.82 |
| | 2014 | 158,739.3 | 69.66 | 7471 | 3.28 | 48.59 | 89.69 | 48,421.35 | 0.78 |
| | 2019 | 170,139.6 | 74.66 | 6291 | 2.76 | 53.03 | 70.80 | 61,633.45 | 0.80 |
| Grassland | 2004 | 480.78 | 0.21 | 1806 | 0.79 | 0.01 | 46.10 | 606.65 | 0.37 |
| | 2009 | 1884.42 | 0.83 | 5520 | 2.42 | 0.02 | 86.82 | 778.03 | 0.40 |
| | 2014 | 374.13 | 0.16 | 1025 | 0.45 | 0.01 | 34.27 | 831.81 | 0.47 |
| | 2019 | 163.62 | 0.07 | 730 | 0.32 | 0.00 | 29.51 | 510.71 | 0.31 |
| Farm land | 2004 | 94,626.63 | 41.52 | 6496 | 2.85 | 35.70 | 108.75 | 33,196.95 | 0.82 |
| | 2009 | 74,865.06 | 32.85 | 7628 | 3.35 | 27.20 | 126.88 | 22,366.56 | 0.79 |
| | 2014 | 63,840.24 | 28.01 | 11276 | 4.95 | 16.49 | 148.65 | 12,902.39 | 0.76 |
| | 2019 | 51,924.33 | 22.78 | 8765 | 3.85 | 7.18 | 137.17 | 13,500.61 | 0.77 |
| Built-up area | 2004 | 2988.72 | 1.31 | 3250 | 1.43 | 0.19 | 65.30 | 2095.70 | 0.64 |
| | 2009 | 2964.24 | 1.30 | 3546 | 1.56 | 0.14 | 72.34 | 1904.96 | 0.60 |
| | 2014 | 4270.86 | 1.87 | 6759 | 2.97 | 0.53 | 91.01 | 1440.06 | 0.58 |
| | 2019 | 4476.42 | 1.96 | 7387 | 3.24 | 0.46 | 96.96 | 1381.03 | 0.56 |
| Water bodies | 2004 | 104.04 | 0.05 | 187 | 0.08 | 0.02 | 11.46 | 1268.00 | 0.68 |
| | 2009 | 375.93 | 0.17 | 475 | 0.21 | 0.04 | 23.36 | 1803.55 | 0.65 |
| | 2014 | 247.86 | 0.11 | 359 | 0.16 | 0.04 | 18.33 | 1573.37 | 0.66 |
| | 2019 | 1153.98 | 0.51 | 3703 | 1.62 | 0.04 | 64.53 | 710.11 | 0.43 |
| Unused land | 2004 | 1752.30 | 0.77 | 8496 | 3.73 | 0.01 | 101.41 | 469.92 | 0.27 |
| | 2009 | 46.53 | 0.02 | 87 | 0.04 | 0.00 | 14.13 | 1218.77 | 0.39 |
| | 2014 | 420.57 | 0.18 | 946 | 0.42 | 0.00 | 34.56 | 1013.21 | 0.50 |
| | 2019 | 35.01 | 0.02 | 37 | 0.02 | 0.01 | 6.40 | 2156.32 | 0.71 |

In general, the maximum patch index and mean patch area increased for woodland and grassland LULC types, despite a slight decrease in the last quarter of the study period. Whereas farmland and unused land followed an inverse trend. The built-up area slightly increased in the first quarter (2004–2009), decreased in the second quarter (2009–2014) and stabilized for the rest of the period (2014–2019), considering the maximum patch index (Table 8). However, concerning the mean patch area, a consistent trend of increment during all period was observed. The water bodies showed an oscillatory trend during the study period for both maximum patch and mean patch area indices.

*3.3. Ecosystem Services Value*

Our results on the ecosystem services valuation showed that the ESV tended to increase during the past 15 years, with an overall increase of $10016.1 \times 10^5$ yuan, showing an equivalent increment of 27.8%. (Table 9). On the one hand, from the perspective of the annual change rate, this upward trend still not relevant for the landscape dynamics along the years evaluated. On the other hand, the ESV of all landscape types increased from 2004 to 2019, except for farmland, grassland, and unused land in which the overall values declined. Among the assessed landscape types, the ESV of woodland had an increment of

32.9% on its area, which equivalent to 10462.2 $\times$ 10$^5$ yuan. The value of water ES increased by 1009.1%, equivalent to 1424.7 $\times$ 10$^5$ yuan. The ESV of grassland decreased by 65.9% and represents 17.2 $\times$ 10$^5$ yuan, which was also considered the largest decrease among all landscape types during the studied period. However, the ESV of farmland decreased by 45.1% equivalent to 1849.8 $\times$ 10$^5$ yuan. The ESV of unused land was the most decreased, reaching 97.8%, and showing an equivalent decline of 3.7 $\times$ 10$^5$ yuan.

**Table 9.** Value and change of ecosystem services in the landscape types from 2004 to 2019.

| Landscape Type | ESV/$\times$10$^5$ (RMB/a) | | | | 2004–2009 | | 2009–2014 | | 2014–2019 | | 2004–2019 | |
|---|---|---|---|---|---|---|---|---|---|---|---|---|
| | 2004 | 2009 | 2014 | 2019 | Change ($\times$10$^5$ Yuan) | Rate % | Change ($\times$10$^5$ Yuan) | Rate % | Change ($\times$10$^5$ Yuan) | Rate % | Change ($\times$10$^5$ Yuan) | Rate % |
| Woodland | 31,719.7 | 36,632.7 | 39,355.5 | 42,181.9 | 4912.9 | 15.4 | 2722.8 | 7.43 | 2826.4 | 7.1 | 10,462.2 | 32.9 |
| Grassland | 26.1 | 102.6 | 20.3 | 8.91 | 76.4 | 291.9 | −82.2 | −80.1 | −11.4 | −56.2 | −17.2 | −65.9 |
| Farmland | 4099.1 | 3243.1 | 2765.5 | 2249.3 | −856.06 | −20.8 | −477.5 | −14.7 | −516.1 | −18.6 | −1849.8 | −45.1 |
| Water bodies | 141.1 | 510.1 | 336.3 | 1565.8 | 368.9 | 261.3 | −173.7 | −34.0 | 1229.5 | 365.5 | 1424.7 | 1009.1 |
| Unused land | 3.7 | 0.10 | 0.91 | 0.08 | −3.6 | −97.3 | 0.8 | 810.0 | −0.8 | −91.21 | −3.7 | −97.8 |
| Total | 35,990.06 | 40,488.6 | 42,478.6 | 46,006.1 | 4498.5 | 12.5 | 1990.0 | 4.9 | 3527.5 | 8.30 | 10,016.1 | 27.8 |

### 3.4. Changes in Individual Ecosystem Services Value

Table 10 presents the changes on the ESV of each service, showing that the food production values experienced the largest decline, with a decrease of 251.8 $\times$ 10 yuan, corresponding to 19.76% of their spatial representativeness, while the largest increase in value was for biodiversity, with an increment of 1065.31 $\times$ 10 yuan (30.7%). A higher change was observed from 2004 to 2009. This change was observed for all individual services, among which the value of water conservation service increased up to 3192.1 $\times$ 10 yuan. While in contrast, from 2004 to 2019, the variation trend of the value of every single function was weakened. After 2014, the change rate of the value of each single ES was the lowest and the fluctuation was the least. From the perspective of the overall individual composition of the research area, the values of each ecosystem service can be ranked as soil formation and protection > water conservation > biodiversity conservation > gas regulation > climate regulation > waste treatment > raw materials > entertainment culture > food production. Values of soil formation and protection services were the maximum, accounting for 15.7% of the complete ecosystem service values, followed by the service of water conservation, accounting for 45.9% of the overall service values. Overall, the biodiversity conservation, gas adjustment and climate regulation services, when combined, accounted for 79.4% of the total ecosystem service values, in which food production was responsible for the overall proportion of 19.7% of the values.

### 3.5. Sensitivity Analysis

All landscape types of ESV had a CS lower than 50%, except for the woodland class (Figure 4). It can be observed that among all landscape classes, woodland showed the highest CS from 2004 to 2019, with a maximum sensitivity index higher than 0.51, which was increased year by year. The CS of each landscape type in the study area was lower than 1, indicating that the CS value adopted is inelastic and applicable to the real state of the study area, suggesting that the estimated results of ecosystem service values are credible.

**Table 10.** Changes in the ecosystem service value (ESV) during the 15 years of study period (2004–2019) in Long County, Shaanxi Province, China.

| Ecosystem Services | ESV/×10⁵ (yuan/a) | | | | 2004–2009 | | 2009–2014 | | 2014–2019 | | 2004–2019 | |
|---|---|---|---|---|---|---|---|---|---|---|---|---|
| | 2004 | 2009 | 2014 | 2019 | Change (×10⁵ Yuan) | Rate % | Change (×10⁵ Yuan) | Rate % | Change (×10⁵ Yuan) | Rate % | Change (×10⁵ Yuan) | Rate % |
| Gas conditioning | 3688.0 | 4019.1 | 4187.4 | 4374.7 | 331.13 | 8.98 | 168.35 | 4.19 | 187.30 | 4.47 | 686.78 | 18.62 |
| Climate regulation | 9361.3 | 1070.3 | 11,406.2 | 12,179.8 | 1341.67 | 14.3 | 703.27 | 6.57 | 773.54 | 6.78 | 2818.48 | 30.11 |
| Water conservation | 6947.6 | 8218.9 | 8592.28 | 10,139.7 | 1271.22 | 18.3 | 373.38 | 4.54 | 1547.51 | 18.0 | 3192.11 | 45.94 |
| Soil formation and protection | 4720.1 | 5079.3 | 5259.76 | 5461.14 | 359.17 | 7.61 | 180.41 | 3.55 | 201.38 | 3.83 | 740.96 | 15.70 |
| Waste disposal | 500.10 | 594.18 | 619.36 | 739.62 | 94.08 | 18.8 | 25.18 | 4.24 | 120.26 | 19.4 | 239.52 | 47.89 |
| Biodiversity conservation | 3469.9 | 3973.7 | 4231.60 | 4535.27 | 503.78 | 14.5 | 257.86 | 6.49 | 303.67 | 7.18 | 1065.31 | 30.70 |
| Food production | 1271.1 | 1155.6 | 1086.06 | 1019.96 | −115.52 | −9.09 | −69.56 | −6.02 | −66.10 | −6.09 | −251.18 | −19.76 |
| Raw materials | 1322.0 | 1380.7 | 1408.84 | 1440.56 | 58.70 | 4.44 | 28.06 | 2.03 | 31.72 | 2.25 | 118.48 | 8.96 |
| Entertainment culture | 1530.0 | 1753.2 | 1865.23 | 2005.94 | 223.26 | 14.5 | 111.97 | 6.39 | 140.71 | 7.54 | 475.94 | 31.11 |
| Aggregate | 32,810.4 | 36,877.9 | 38,656.8 | 41,896.8 | 4067.48 | 12.4 | 1778.9 | 4.82 | 3240.01 | 8.38 | 9086.41 | 27.69 |

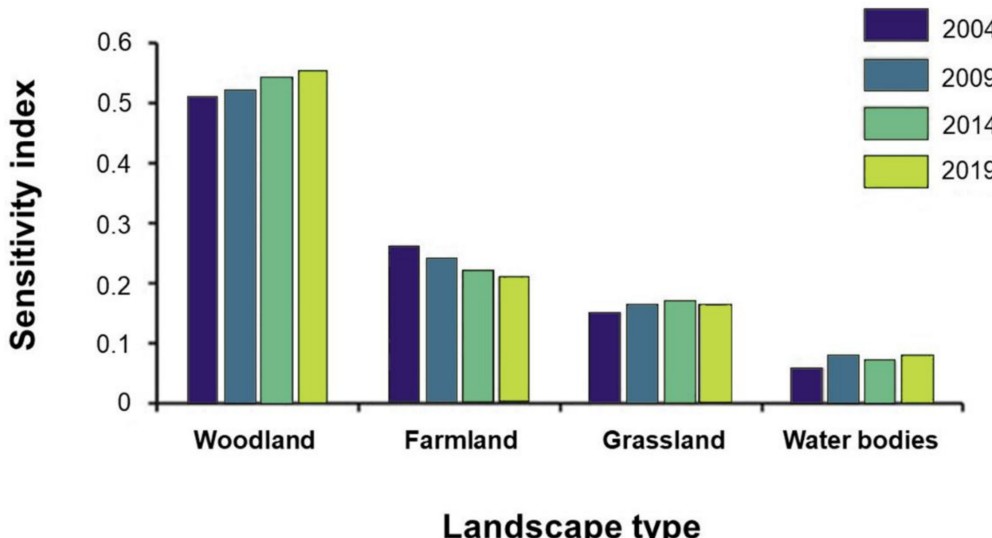

**Figure 4.** Changes on the sensitivity index for the landscape types from 2004 to 2019.

## 4. Discussion

Our findings showed an overall improvement in the spatial distribution of the landscape types and ES evaluated during the 15 years of the study period. With woodland and grassland accounting most for the improvement, whereas the transformation of farmland and grassland into built-up area reduced the landscape cover, ES provisioning and amenity. The results are in accordance with some other studies conducted in China [30,65]. The observed pattern over urban areas can be attributed to increasing middle-class demands in China, which is directly correlated to housing building in cities, hence stimulating the construction booming [59]. In fact, in countries like China, farmland is not only primarily targeted for agriculture, but also housing [25], as they're also better conditions for water supply and settlement patterns.

Overall, evenness and connectivity in Long County improved over the last 15 years, which can be observed by the decline of fragmentation and the degree of landscape dominance, also resulted from an improvement of the patches and the maximum patch

index, which lead the enhancement of spreading degree, Shannon diversity and Shannon evenness index here observed. This improvement in landscape and biodiversity indicators can be mostly attributed to the growth of woodlands and grasslands during 2004–2019, especially in the first third (2004–2009) (see Figure 3; Table 6). These kinds of landscape transformations can alter the species-area relationships, which represent complementary habitats for several species and can support ecological connectivity, thus contributing to the biological community structure [66]. This general pattern can also be highlighted by the higher increment on the sensitivity index for woodland, as well as the gradual increase on the LULC during the study period.

Most of woodlands and grasslands were converted from the farm, water and unused lands. That is a good signal, in the sense that one can infer that this efficient allocation undoubtedly improved or will improve most of the provisioning services (e.g., nutrition, water provisioning and biomass energy) and regulating and maintenance (e.g., regulating wastes, flow, physical and biotic environment). Based on the ESV, the overall value of ES increased (4.34%), with woodland and grassland contributes to this gain. Those results are also in line with the proposed by Zhang et al. [16], who found that the implemented payments for ecosystem services (PES) programs by the Chinese government, notably the Conversion of Cropland to Forest Program (CCFP) and the Ecological Welfare Forest Program (EWFP), provided poor rural farmers with large areas of forest with sizable cash subsidies, which reduce their motivation to migrate and to transform the forest into croplands.

The conversion of farmland, grassland and water yield into built-up area is inevitable in a fast-developing economy, with high population density and fast growth middle-class demands. For instance, a related study showed that the considerable urban expansion and consequent growth in GDP in four ecoregions in the Hang-Jia-Hu region (China), led to a loss of 8.5 billion RMB yuan ESV per year on average between 1994 and 2003 [25]. Another previous research in China also observed that the ratio of ESV per capita to the GDP per capita was about 0.87, and such ratio was lowest in the most economically developed and densely populated areas [31]. Thus, the conversion of forest and agricultural land into urban areas must be done through less environmental impact. For instance, by taking into account the new standards of environmentally friendly cities and smart cities, which also prioritizes the use of clean energies, the construction of buildings that are energy efficient and provide the expansion of green areas [2]. Here, we have noticed that there was a significant unused land that was converted into woodland and grassland, while simultaneously other woodland and grassland were converted into built-up area. It also would be more beneficial for ES enhancement, transferring unused land into built-up area while maintaining the forest land.

The overall enhancement on the ESV was mostly due to the increase on the woodland, grassland and water body with total accounted for 4.34% of the ESV, equivalent to $64.873 \times 10$ yuan, during the study period. A previous study found a decrease of 231.3 million Yuan from 1996 to 2004 in Shenzhen, one of the fastest-growing metropolitan areas in China, mainly due to the decreasing areas of woodland and water bodies [14]. While in the present study, the increase of ESV was mostly due to the transformation of farmland into grass and woodland. It is important to notice that, farmland can sometimes provide more valuable ES depending on the agricultural system and the type of crops that are allocated [24]. For this specific case, the conversion of one unit of farmland into grass or woodland add more value into the ESV, than the opposite. It means that the type of agriculture system and their resulting output need to be improved into more environmentally friendly practices, by understanding the optimal trade-off between the transformation of grassland and woodland into farmland and other land uses, such as water body. However, the local and central government plays a key role in identity and promote environmentally friendly agriculture production by proved the required incentives that internalize the positive externalities while externalizing negative externalities.

Despite the observed increment on the ESV of grassland, it is important to highlight that grassland is a fire-prone ecosystem that can easily burn and spread faster, hence results on the release of greenhouse gases and the subsequent soil impoverishment and the degradation of macrofauna [66]. In such circumstances, the relative gain on ESV can be easily converted in disservices. In that regarding, the government need to ensure that most of the grassland can be converted into woodland, before fire occurrence, since there is considerable potential for this conversion, based on the transition matrix (see Table 6).

ESV increased during the study period (2004–2019) because most of the farmland (agriculture) was converted into woodland and grassland. The enhancement of ESV can also be attributed to government policy, that aim to restore forest landscape and its environmental management to prevent soil erosion and habitat loss [16,19], since areas with larger slope are not suitable for farming, which shall be converted to woodland and grassland [21]. Most of the unused land was converted into water bodies, thus this land class was efficiently converted. As a result, the overall ES was gradually improved. With human intervention, the proportion of patch area and the maximum patch index of built-up area gradually increase over the years, while the aggregation degree of water and the landscape shape index decreased, indicating that under the intervention of anthropogenic activities, the degree of water conservation in the area is higher, while the unused land is maximized.

The values of soil formation and protection service were the highest, followed by the service of water conservation, biodiversity conservation, gas adjustment and climate regulation service function. While food production accounted for the lowest proportion of the overall service value. The main purpose is that the increase of woodland and grassland improves the regional ecosystem functioning, which directly affected the services of soil formation and protection, and water conservation. Therefore, the food production capacity is weakened, resulting in a decline in the value of food production services in the region.

## 5. Conclusions

We validated the benefits of using LULC from fused satellite imageries in evaluating land use changes on ES and their economic values in fast-developing county like Long County. Our findings showed that the overall spatial patterns of LULC were improved during the study period, especially from 2014 to 2019, with the conversion of farmland and unused land into woodland and grassland accounting for most of this improvement. This suggests improved conservation outcomes of landscape connectivity, uniformity and biodiversity in that period. The total ESV in the study area was also higher compared to the national average, despite the expansion of the built-up area, which was probably related to the existing of government policies that stimulating the enhancement of ES by providing a financial incentive.

We also examined the landscape pattern information and characteristics of its structure and spatial configuration between 2004 and 2019 in Long County. Even though this research didn't establish a direct relationship between pattern and ESV, we provided landscape pattern evidence at the patch level, landscape-level, and class-area level, which is suitable for quantitative expression of the relationship between landscape pattern and ecological process. We particularly recommend studying the relationship between landscape pattern and ESV in the future.

The analysis conducted in this study highlights the importance of understanding the causes of drivers and underlying drivers of LULC change to assess the impacts of those changes and define development policies. The implementation of development models at the landscape level must balance the economic benefits and ecological gains of the different land cover classes to enhance their ESV, requiring an interdisciplinary and science-based approach.

**Author Contributions:** Conceptualization, W.S., R.P., M.S. and A.A.M.; methodology, W.S., A.A.M. and R.P.; formal analysis, W.S., M.S. and R.P.; investigation, W.S., R.P. and A.A.M.; writing—original draft preparation, W.S., R.P. and M.S.; writing—review and editing, W.S., R.P., J.M.N.S., P.C. and F.S.C.; supervision, R.P., P.C. and F.S.C.; All authors have read and agreed to the published version of the manuscript.

**Funding:** This study was supported by the Research on Capitalization of Natural Resources and Corresponding Market Construction in China [grant number 15ZDB162]; and partially through the FCT (Fundação para a Ciência e a Tecnologia) under the projects PTDC/CTA-AMB/28438/2017-ASEBIO and UIDB/04152/2020—Centro de Investigação em Gestão de Informação (MagIC). This research was also funded by the Forest Research Centre, a research unit funded by Fundação para a Ciência e a Tecnologia I.P. (FCT), Portugal (UIDB/00239/2020).

**Institutional Review Board Statement:** Not applicable.

**Informed Consent Statement:** Not applicable.

**Data Availability Statement:** Samples of data are available from the authors.

**Acknowledgments:** The authors would like to thank the financial support provided by Research on Capitalization of Natural Resources and Corresponding Market Construction in China and Fundação para a Ciência e Tecnologia (FCT).

**Conflicts of Interest:** The authors declare no conflict of interest.

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
