# Peer review of "Using Satellite Image Fusion to Evaluate the Impact of Land Use Changes on Ecosystem Services and Their Economic Values"

_remotesensing, doi:10.3390/rs13050851_

Round 1

Reviewer 2 Report

Please find review in the attached file.

Reviewer 3 Report

Brief summary

This work presents a use of RS images, fused from 2 different sensors, in assessing the impact of land use and land cover (LULC) changes on ecosystem services (ES) and their economic values over 15 years (2004-2019) in the Long county, Shaanxi province, China.

The article is well structured and contains many analyses. Some aspects should be better described or extended in order to guarantee a better understanding of authors' approach. Several problems/doubts were identified and summarized in the following general comments and in the Specific comments section.

The manuscript does not bring any novelty to ecosystem services value assesment or LULC change detection with satellite data.

General comments

  1. Using fused images – are they needed in the research? LISS III (23 m) are not high resolution images, like the authors stated (ln 159-160). They are the medium resolution, very the same as Landsat images (30 m). Were there the benefits of using fused images, and what they are?

    Despite what the authors stated, that is “We walidated the benefits of using fused satellite imageries from the multiple sensors in evaluating land use changes on ES and their Economic Values ...” (ln 431-432), there is no such a validation (or comparison to using single images) presented in the manuscript.

  2. 2.7. Evaluation of Landscape pattern: 1) Are LULC changes, calculated based on landscape indexes, needed for something in the research? These changes weren't presented in relation to ecosystem services changes. 2) What were the criteria of the landscape indicators selection used in the research? Some of them is also redundant/reducible. 3) The authors used 7 indexes. Interpretation of some of them (like CA or NP) is easy. But what about rest of them, what their changes indicate indeed? 4) Use one name (one among: “landscape indexes”, “landscape indicators”, “landscape metrics”) throughout the text, not three.

  3. Assessment of ecosystem services values explained better then in other articles (for example [14]) but still not clearly enough, and requires further working. See specific comments for details.

  4. Sensitivity analysis (SA): No theoretical foundation for the need of SA usage is presented. All references provided in section 2.9 are not appropriate (a cited publication does not assist in judging the correcteness of authors' statement), which causes difficulty in recognition the correctness of this step of used methodology. The description in the section 3.5 is not clear.

  5. The major doubt concerning acceptance of the manuscript relates to the fact that this work is similar to other research (some of them are cited in the manuscript), i.e. it uses the same methods and materials but for other time period and study area (geographical location). The manuscript was submitted to Remote Sensing journal, so there is natural need to ask the question “what issue's understanding related to RS does increase this work”.

    This work is a good analysis for the Long county municipal council. Maybe the proper journal for the manuscript would be those published researches from the fields like land management or land development or spatial economy or sustainable development.

Specific comments

  1. Title: requires revision, in my opinion.

  2. ln 22-24: The sentence requires revision, because the Coefficient of Sensitivity (CS) analysis was used not for assessing application of several landscape metrics but "To verify whether the selected Economic value (VC) was suitable for the study area ..." (ln 231).

  3. ln 33-34: The sentence requires revision.

  4. ln 93-94: The sentence requires improvements.

  5. ln 104-105: The sentence is unclear.

  6. ln 112-113: The sentence is unclear.

  7. Figure 2: The tilte requires correction.

  8. Figure 2: I suggest to change “ Rotation Forest Classification ” to “ Classification with Rotation Forest method ” or “ 'Rotation Forest' calssification ”.

  9. Figure 2: Does ecosystem services value assesment come from Landscape pattern analisis, or it was done based on LULC data?

  10. Section 2.4: What was the target image or what was the source of GCPs used in geometric correctoin of Landsat and IRS images?

  11. ln 142: Citation [34] is not appropriate in the sentence.

  12. ln 155: Citation [40] is not appropriate in the sentence.

  13. ln 169: "construction land" – is it good/appropriate name of the class? What about using "urbanized area" or “built-up area”?

  14. ln 173: The description of step (i) is not clear.

  15. ln 175: What the PCA was used for in the research?

  16. ln 183-184: What does “... the extracted random values were recognized from GEE images ...” mean, it's not clear?

  17. ln 186: Citation [44] is not appropriate in the sentence.

  18. ln 186: Is ERDAS Imagine was used only to "examine the accuracy of classified LULC maps", and - if yes - why?

  19. ln 188-189: Acceptabel accuracy of classification dosn't mean that "maps are similar to the reference data".

  20. ln 189: Citation [44] is inappropriate in the sentence.

  21. ln 195: Providing only citation [25] is not enough while such basic features of landscape indexes are pointed out.

  22. ln 199: What does “class-area level” mean? Is it patch-type level?

  23. ln 200: Citation [13,25] is inappropriate in the sentence.

  24. ln 201: The "patch number (NP)" - incorrect name.

  25. ln 202: "CLUMPY" – reqiures description.

  26. ln 205: The "landscape richness (PR)", “Shannon evenness (SHEI), spread degrees (CONTAG)“ - are incorrect names and have to be changed.

  27. ln 207: Citation [13,46] is inappropriate in the sentence.

  28. Table 3: Check very carrefuly the names in the first column and description in all table and correct where required.

  29. Table 3: What was the source of the information contained in the table?

  30. ln 220, 226: It is not possible to find and read the article [30] in english. As the result it's not possible to check the usage of the formula (2) and (3).

  31. ln 221: The article [49] dosn't contain the (2) formula. Was this formula prepared by the authors?

  32. Equation (2): Was it used for all land use classes? What about using of P (typical grain price) in case of VC calculation for example for such LU classes as water, woodland or construction land – grain isn't grow there?

  33. Equation (2): In my opinion it's required to present VC values used in the research for each land use classes. Also, was the P (typical grain price) value the same for every analysed year/period?

  34. ln 222: What is “a-1”?

  35. ln 224: The “n is the annual yield” requires attantion and maybe correction.

  36. ln 232: The [50] literature presents the use of SWAT model and, among other, sensitivity analysis for it. The authors don't use SWAT model, so the citation is inappropriate. Some other literature, which directly describes sensitivity analysis, maybe found and provided.

  37. ln 232-234: The sentence requires correction.

  38. ln 234: Citation [7] is completely inappropriate here.

  39. ln 236: (Zhang et al. 2017) isn't provided in the References section.

  40. ln 237 Citation [31] is inappropriate.

  41. ln 239: Citation [7] is inappropriate.

  42. ln 242: A part of the sentence requires corection of English.

  43. ln 242: “k stands for a different landscape” - is this true, not for different LU class.

  44. Figure 3: I suggest to increase the sizes of map images and decrease the size of the legend.

  45. Table 4 and 5: are not easily readable – I suggest to change font size where appropriate.

  46. Table 5: I don't understand what describe the numbers in the first 7 lines of the table or where this data come from.

  47. Table 5: The title of the first comumn of the table should be “Landscape types in 2004”.

  48. ln 258-260: The sentence requires clarification.

  49. ln 269: Less then 1% change in 15 years cannot be called “increase”.

  50. ln 407-408: Woodland is also fire-prone.

  51. ln 410-411: The sentence requires correction.

  52. reference no 27: Correction required.

  53. reference no 38: Correction required.

  54. All references positions should be carefully checked and completed or corrected where required.

Round 2

Reviewer 1 Report

All my questions have been answered.

Author Response

Thank you very much for considering our work.

Reviewer 2 Report

The authors addressed the main concern of my review, i.e. they provided the full classification validation results. These results also highlighted the importance of using fused imagery.

I have still few doubts about the atmospheric correction procedure, but the validation shows that there is adequate accuracy.

The authors also gave detailed answers to all reviewers.

Taking into consideration the above, I suggest that the manuscript can be published as it is.

Author Response

(The authors gave the same response as above.)
